# FORTIFYING HALLUCINATION DETECTION TO OUT-OF-DOMAIN DATA

## ABSTRACT

Hallucinations remain one of the major barriers to the reliable deployment of Large Language Models (LLMs). Recent works have explored both supervised classification based approaches and unsupervised metric based approaches, with the latter remaining popular since they do not require labeled data. However, unsupervised methods lag behind supervised ones for in-domain data, despite having slightly better performance out of domain, as we show across 11 datasets and 10 models. This underscores the importance of supervised approaches, but also highlights their weakness in generalizing to unseen domains. To narrow this generalization gap, we introduce a simple approach to make supervised hallucination detectors more generalizable by relying on a curated, multi-domain training mix, which can complement subsequent addition of task-specific data. In our experiments on hallucination detection on 697K QA samples from 12 open source QA datasets, we show that incorporating this general training allows supervised methods to surpass unsupervised metric based methods by an average of +7.25% on out of domain data, without addition of any task-specific data. We also analyze scaling behaviors and estimate how much task-specific data is required to achieve reliable performance, finding that models augmented with general data require up to 40.3% less task-specific data to achieve close to optimal performance. Together, our findings highlight both the brittleness of existing supervised hallucination detectors and a simple path toward fortifying them detection against domain shift.

## 1 INTRODUCTION

Large language models (LLMs) are increasingly deployed across a wide range of applications, however their tendency to generate factually incorrect or misleading outputs, often termed *hallucinations*, poses a critical barrier to their adoption in high-stakes or out-of-distribution settings (Kim et al., 2025; Dahl et al., 2024). Detecting hallucinations at test time is challenging (Sahoo et al., 2024), a naive approach is to fact-check outputs against an external reference or database (Chern et al., 2023; Min et al., 2023). However this requires costly retrieval and fails when no ground-truth reference exists, calling for other methods to detect hallucinations.

One line of research has leveraged *uncertainty estimates* as potential signals of model reliability, and thus, for hallucination detection (Farquhar et al., 2024). *Unsupervised* methods analyze output probabilities, response consistency, or cluster semantic entropy (Abdaljalil et al., 2025; Farquhar et al., 2024; Nikitin et al., 2024; Venhuizen et al., 2019), or use linear probing (Kossen et al., 2024), while *supervised* methods train classifiers on model states to detect hallucinations (Liu et al., 2024) (Figure 1). While supervised methods outperform unsupervised metrics in-domain (Liu et al., 2024), they lose robustness out-of-domain, often underperforming unsupervised methods when faced with distribution shift. We empirically validate this in §4.1 by benchmarking supervised against unsupervised approaches across multiple data domains and model families, showing that while supervised methods excel in-domain, their performance degrades sharply under domain shift, often under-performing unsupervised methods.

Motivated by this, we introduce an approach to bridge this performance-generalization gap by training a supervised model on a large and heterogeneous dataset to yield more robust hallucination detection. Inspired by domain generalization and pretraining works that show how diverse training data can improve out-of-domain robustness in other tasks such as NLI, NER and sentiment analysis

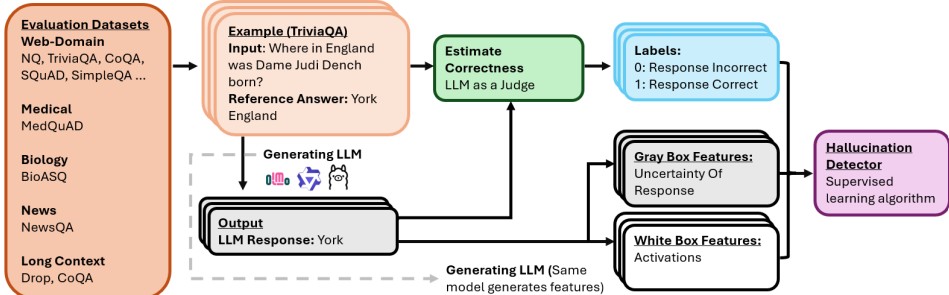

Figure 1: Overview of supervised learning based hallucination detection pipeline. We use examples from QA style benchmarks to generate positive and negative examples of when LLM an hallucinates. Along with generated features based on the prompt and LLM response, a downstream supervised learning algorithm is applied to train a classifier that detects these hallucinations based on the input features. We have two types of data, a general training mix and a task specific set, both of which we use to train the model, the task specific set confers in-domain specificity while the general training mix works to prevent overfitting, promoting robustness under domain shift.

(Hosseini et al., 2024; Stacey et al., 2025; Yu et al., 2022), we adapt these principles to uncertainty-based hallucination detection. In addition, we study scaling law type behaviors, investigating how much domain-specific data is needed to obtain a classifier with acceptable performance. We conduct our experiments using different dataset splitting strategies, iteratively testing combinations of train-test splits across a comprehensive dataset of 697K QA samples from 12 open source QA datasets, which we further categorize into broad domains. Our evaluation spans 10 different LLMs of varying sizes to ensure robustness across model families.

**Our main contributions are:**

- We present a comprehensive benchmarking of supervised hallucination detection methods against state-of-the-art unsupervised metric-based approaches, across a wide range of domains and language models, finding that unsupervised methods outperform supervised ones out-of-domain but not in-domain.

- We introduce a data-driven strategy for improving out-of-domain robustness of supervised hallucination detectors by scaling up heterogeneous training data, demonstrating its effectiveness at narrowing the performance-generalization gap.

- We provide a systematic analysis of the relationship between training data volume and classifier performance, offering practical guidelines on data requirements for effective uncertainty-based hallucination detection.

## 2 BACKGROUND & RELATED WORKS

**Background: Hallucination Detection via Uncertainty Quantification**  Broadly, there have been two styles of approaches for uncertainty quantification. On one hand, *unsupervised* methods generally seek to estimate uncertainty via analyzing output probabilities, consistency and similarity across multi-generations (Abdaljalil et al., 2025; Farquhar et al., 2024; Nikitin et al., 2024) and via linear probing (Kossen et al., 2024). For example, semantic entropy (Venhuizen et al., 2019) clusters multiple generated responses to estimate semantic entropy, where prompts leading to diverse sets of clusters implies a higher level of uncertainty. On the other hand, work has explored *supervised* classifiers with internal language model states as features trained to estimate uncertainty (Liu et al., 2024). Figure 1 demonstrates this approach in the white-boxing setting, where internal states of the LLM are used to predict labeled hallucinations. This approach also applies to black-box and gray-box settings. For example, we can restrict ourselves to the use aggregated statistics based on the output logits of the generating LLM in the gray-box setting. In addition, when combined with a supervised model that produces probability estimates, these probabilities can be interpreted as uncertainty estimates.

**Benchmarking UQ and Hallucination Detection**   Previous efforts to benchmark uncertainty quantification have focused on examining performance vs. efficiency (in terms of number of generations needed) tradeoffs of UQ methods requiring multiple generations (Xiong et al. (2024); Valentin et al. (2024)). Ye et al. (2024) utilizes UQ metrics and various datasets to benchmark the certainty level of different LLMs and calibratedness of their responses. Xiong et al. (2023); Tian et al. (2023) provide extensive benchmarks across different datasets and models but focus mainly on verbalized uncertainty, also they do not benchmark on in-domain vs out-domain. Lastly Liu et al. (2024) benchmarks supervised uncertainty quantification methods across methods and studies the extent of domain transfer across datasets, but does not focus on when such domain transfer occurs.

**Out-Of-Domain Robustness**   Approaches to improving out-of-domain robustness in machine learning include training and optimization-based techniques (Wang et al., 2022; Yang et al., 2021) as well as data-centric approaches. In computer vision, increasing diversity of training data and data augmentation have been shown to reduce overfitting and improve generalization (Zhou et al., 2020; Rahman et al., 2019). In NLP, synthetic data have been explored as a means to enhance domain generalization in natural language inference (Hosseini et al., 2024). Building on these insights, we study whether large and heterogeneous pretraining can play a similar role for hallucination detection under domain shift.

## 3   APPROACH & EXPERIMENTAL SETUP

Our goal is to develop a generalizable method for hallucination detection in the question-answering (QA) setting. We adopt a supervised classification approach, showing that it outperforms alternative metric based methods, but that its performance degrades out of domain. To address this challenge, we explore training on a mix of in-domain and heterogeneous data. This section introduces the task setup and supervised detection approach as well as datasets, models and experimental setup.

**Hallucination in QA setting**   While ultimately our goal is to measure hallucinations in any LLM output, as a starting point, we focus mainly on the QA setting, where a LLM generates a textual response to an input question. This response is scored against a reference response, and incorrect responses are treated as hallucinations. The task is then to detect such hallucinations given the input question and corresponding LLM response. We assume a white-box setting with full access to model internals such as activations and output logits.

We view performance in short-form QA as foundational for hallucination detection in longer-form generative tasks such as abstractive summarization. This is evidenced by many long-form methods that first extract atomic facts as a pre-processing step (Thirukovalluru et al., 2024; Min et al., 2023; Kadavath et al., 2022), making fact-level hallucination detection essential. The QA setting is also advantageous as it is relatively straightforward to measure correctness, whereas long-form outputs require fact-extraction and verification across many sentences, a capability that is still an active area of research (Chen et al., 2025; Liu et al., 2025; Wei et al., 2024b).

**Our Approach**   Our approach focuses on training a multi-domain supervised hallucination detector using white-box and grey-box LLM features (Figure 1). The pipeline has three components: (1) generating candidate answers from the *generating LLM* on QA datasets (2) constructing feature representations from their internal activations and (3) training a classifier to distinguish between hallucinated and correct responses. Formally, each sample consists of features $(X_{\text{prompt}}, X_{\text{response}})$ paired with a binary label $y \in \{0, 1\}$, where $y = 1$ denotes a hallucinated (incorrect) response and $y = 0$ a correct one.

For features, we follow prior work (Liu et al., 2024; Azaria & Mitchell, 2023) and extract activations from the middle and final layers of the generating LLM, using only activations corresponding to the last token of both the prompt and generated response. We then apply dimensionality reduction using truncated SVD, necessary because the activations are high dimensional (3584-4096) and training data is limited for some of our splits. Our experiments focus on how data can affect performance, thus for consistency, we apply SVD across all models even when dataset size exceeds the feature dimension.

We evaluate classifiers of the form

$$f(X_{\text{prompt}}, X_{\text{response}}) = y_{\text{score}}, \quad y_{\text{score}} \in [0, 1], \tag{1}$$

where $y_{\text{score}}$ denotes the predicted score of hallucinations, with high scores indicating a higher likelihood of hallucination. Final predictions are obtained by thresholding $y_{\text{score}}$, although we also evaluate quality of the score itself as described in §3.2. In our experiments, we use a Random Forest classifier, following prior work in Liu et al. (2024). Random Forests are widely regarded as a strong and robust baseline across many different problem settings (Wainer, 2016), offering good performance with minimal tuning (Probst et al., 2019). We also experimented with other classifiers, namely XGBoost (Chen & Guestrin, 2016) and penalized linear regression and found Random Forest to be very competitive with these. Additionally we keep hyperparameter constant throughout all experiments (Appendix E). While optimal hyperparameter vary across dataset splits, we adopt a single setting to avoid exhaustive tuning, further our focus is on in-domain vs out-domain performance which we find to be largely insensitive to hyperparmeter choice.

## 3.1 DATASETS, MODELS AND METHODS

**Evaluation Datasets**   We collect data from 12 different QA style benchmarks across different domains and input context length. These datasets along with their domains are listed in Table 1. Several of the datasets come with a train and a test set. For all our tested models, we found that the performance on both these sets tended to be similar, and thus we use both. Combined this gives us a dataset of 697K examples.

Table 1: Benchmarking Datasets used in our study.

| Benchmark | Citation | Domain | Dataset Size |
|---|---|---|---|
| TriviaQA | Joshi et al. (2017) | Encyclopedic | 76.5k |
| NQ | Kwiatkowski et al. (2019) | Encyclopedic | 91.5k |
| bioASQ | Tsatsaronis et al. (2015) | Biology | 4.7k |
| CoQA | Reddy et al. (2019) | Conversational | 116.6k |
| DROP | Dua et al. (2019) | Reasoning-heavy | 83.6k |
| HotpotQA | Yang et al. (2018) | Encyclopedic | 97.8k |
| MedQuAD | Ben Abacha & Demner-Fushman (2019) | Medical | 16.4k |
| NewsQA | Trischler et al. (2016) | News | 78.3k |
| SimpleQA | Wei et al. (2024a) | KB QA | 4.3k |
| SQuAD | Rajpurkar et al. (2016) | Encyclopedic | 98.0k |
| WebQuestions | Berant et al. (2013) | KB QA | 5.8k |
| OpenLLM | Myrzakhan et al. (2024) | Various | 23.7k |

**LLMs Considered**   Our evaluation spans three model families: Llama-3.1 (Dubey et al., 2024), OLMo2 (OLMo et al., 2024) and Qwen2.5 (Team, 2024). Additionally we consider various post training schemes, Tulu 3(Lambert et al., 2024) and SimPO-based (Meng et al., 2024) for Llama-3.1, and the post training process in (Mu et al., 2025) for Qwen-2.5. We evaluate models across different training stages, namely SFT and SFT+DPO. For computational purposes, we primarily use the 7-8B variants of these models except for OLMo2 where we considered both 7B and 32B models.

**Baseline Unsupervised Methods**   We consider a mix of multi generation and single generation methods. These are Semantic Entropy (Farquhar et al. (2024); Venhuizen et al. (2019)), Sindex (Abdaljalil et al. (2025)), GNLL (Aichberger et al. (2024)) and PTrue (Kadavath et al. (2022)). We elaborate more on these methods in Appendix A.

## 3.2 EVALUATION METRICS

We evaluate performance using two metrics. For threshold-agnostic evaluation, we report Area Under Receiver Operating Curve (AUROC), which measures how well a score differentiates between positive and negative examples across all decision thresholds. AUROC has been used extensively

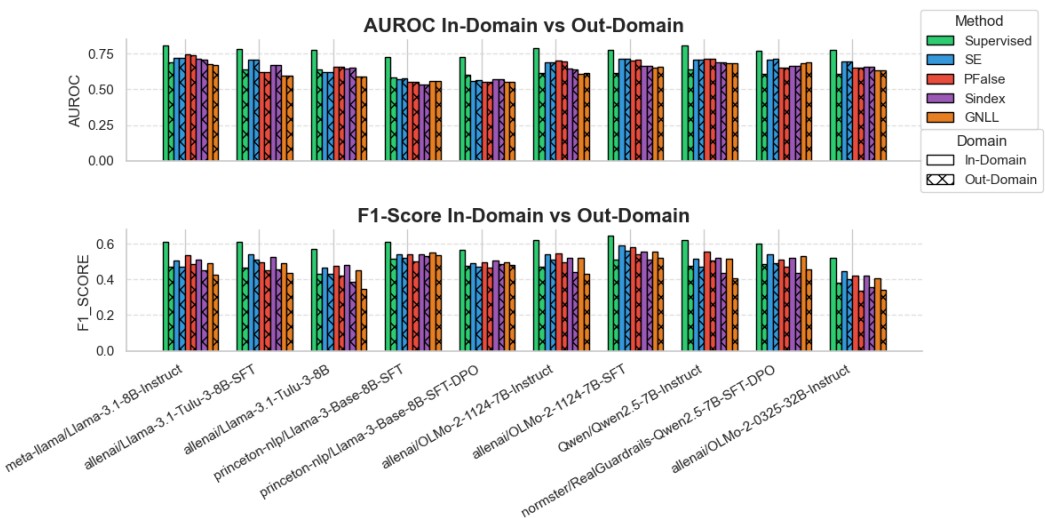

Figure 2: Aggregated In-domains and Out-of-domain evaluations for various models macro-averaged across 11 different datasets. Solid bars represent in-domain performance and cross-hatched represents out-domain performance. Top Row: AUROC, Bottom Row: F1-Score

in previous work (Liu et al., 2024; Aichberger et al., 2024; Farquhar et al., 2024). For threshold-aware evaluation, we use F1-score, optimizing thresholds on validation data to maximize this metric. Additional details on these and alternative metrics are provided in Appendix B.

### 3.3 LLM As a Judge

We use an LLM as a judge to determine correctness, as prior work has shown that alternative metric-based approaches (e.g. ROUGE,BLUE,BERTSCore) can yield inaccurate labels and substantially alter results (Santilli et al., 2024; Ielanskyi et al., 2025; Janiak et al., 2025). We initially tested `gpt-4o` and gpt-4o-Mini, then sought open-source alternatives for cost efficiency. Among these, `Qwen3-14B` (Yang et al., 2025) showed strong agreement with `gpt-4o` on a large 10K set of samples as well as with human evaluators on a smaller sample of 100 data points. Thus, we adopt `Qwen3-14B` as our primary judge model for labeling hallucinations.

## 4 RESULTS

Towards building a generalizable hallucination detection system we first quantify the degree of supervised performance loss out-of-domain (§4.1). Then, we demonstrate the effectiveness of our generalized heterogeneous dataset to reduce domain gap (§4.2). Finally, we investigate scaling law type behaviors, namely how performance scales with number of data samples (§4.3).

### 4.1 HOW DO SUPERVISED METHODS PERFORM AGAINST UNSUPERVISED METHODS IN BOTH THE IN-DOMAIN AND OUT-OF-DOMAIN SETTING?

We compare supervised and unsupervised methods in both the in-domain and out-of-domain settings across the 11 benchmark dataset. For each source dataset, performance is evaluated on held-out source data (in-domain) and all other datasets (out-of-domain), repeating this for all source-target pairs and generating LLMs. Figure 2 reports these results macro-averaged across datasets. Additionally we report performance gaps and associated error bars that account for dataset-specific variations in Appendix F.

As expected, supervised methods consistently outperform unsupervised ones in-domain across all benchmarks (Figure 2), with this advantage being consistent across model types and evaluation metrics. Out-of-domain, however, supervised models experience substantial performance drops,

often losing their in-domain advantage and performing on par with or even below unsupervised metrics, highlighting their sensitivity to distribution shift. Among unsupervised methods, the multi-generation based metrics, SE, Sindex and PFalse generally outperform GNLL, which uses a single generation. Performance varies across generating LLMs, with SE often the top performer, consistent with prior work (Farquhar et al., 2024) though other studies report different rankings (Abdaljalil et al., 2025), likely due to differences in generating LLM or evaluation setup.

## 4.2 CAN TRAINING ON A LARGE GENERAL DATA-MIX MITIGATE OUT OF DOMAIN PERFORMANCE DEGRADATION?

Next, we explore our primary question of how dataset diversity affects generalization. We first use a dataset-level leave-one-out split, where one dataset is held out as the target, termed the **Task-Specific (TS) dataset**. The remaining data sets are then combined into a larger, more diverse **General (GE) dataset** for training. We further while retain a **Task-Specific (TS) dataset** for fine-tuning, to test whether increased diversity mitigates the drop in performance when moving from in-domain to out-of-domain evaluation.

**Leave-one-out experiments.** We first evaluate our approach using a dataset level leave-one-out data split, for this we iteratively select one dataset to leave out and function as the target dataset. The remaining datasets then function as the general set. The target dataset is split into a train, test and validation dataset, this allows us to assess the hallucination detection capabilities under two settings, 1) where the classifier is trained solely on the GE set 2) where the classifier is trained on both the GE set and some data from the target domain / TS set.

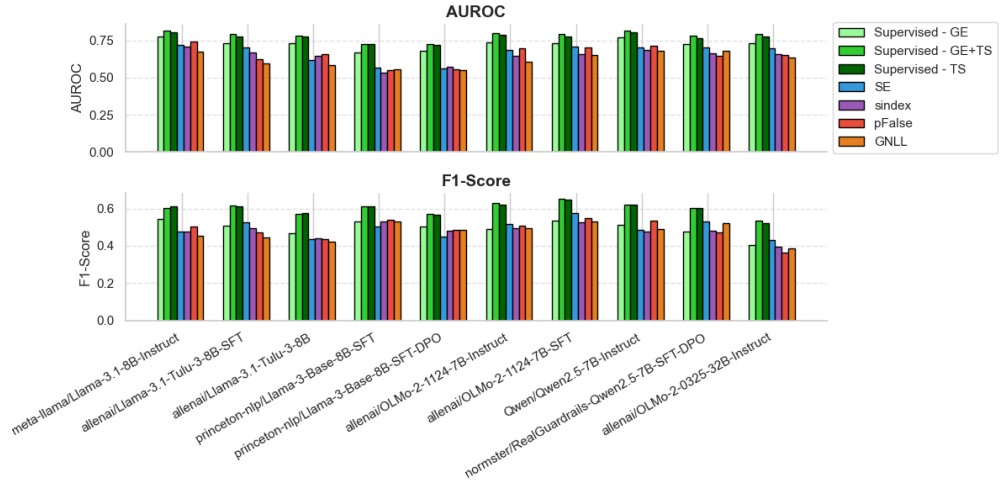

Figure 3: AUROC and F1-Score of different methods aggregated across heldout-benchmarks. Groups represent the detection performance for different generating models. Bars represent the method used with green bars showing different variants of supervised methods (differing by training data used) and remaining bars representing unsupervised methods.

Figure 3, displays results aggregated over the heldout-target datasets for the different LLMs. We highlight three key observations:

*First*, we examine the value of target-specific data. Across all held-out targets and generating LLMs, classifiers trained exclusively on general (GE) data consistently under-perform compared to those trained with task-specific (TS) data, underscoring the important of target-specific supervision. This result is unsurprising given our earlier findings on the effectiveness of supervised methods in-domain. Interestingly, the gap between GE-only and TS-only models persists even for target datasets with strong similarities to others in the GE set, these would include target datasets such as Natural Questions. We have observed in the single source experiments, that in some cases training on another source can outperform a training on the target source. Thus the consistent overall gap suggests a degree of negative transfer when relying on heterogeneous GE data. Nevertheless, when

comparing expected performance of scombingle source on out-of-domain, GE-only still seems to provides a net benefit.

*Second*, supervised GE models generally outperform unsupervised methods. Comparing GE-only to unsupervised baselines, Figure 3 shows that GE-only generally matches or exceeds the best unsupervised methods across models. This advantage is especially pronounced in AUROC, where GE-only achieves higher average performance than all unsupervised approaches. However, the pattern is less consistent for F1-score: for `OLMo2-Instruct`, `Qwen2.5-Instruct`, and `Llama3.1-Tulu-SFT`, GE-only underperforms. A closer analysis for `OLMo2-Instruct` (Figure 4) reveals that this dip stems from poor performance on longer-context datasets DROP and CoQA. Since AUROC remains high, this suggests some sensitivity to thresholding on the general set. Similar patterns are observed for the other models.

*Finally*, GE+TS offers limited benefits over TS when target data is abundant. After averaging across heldout target datasets, we find that GE+TS slightly outperforms TS-only by only a small margin. We hypothesize that target dataset size explains this. In particular the target datasets tested are generally quite large (on the order of tens of thousands of examples). In such cases, classifiers may already be saturated by task-specific data, limiting additional performance gains when adding the general examples. We explore these scaling effects more systematically in later sections.

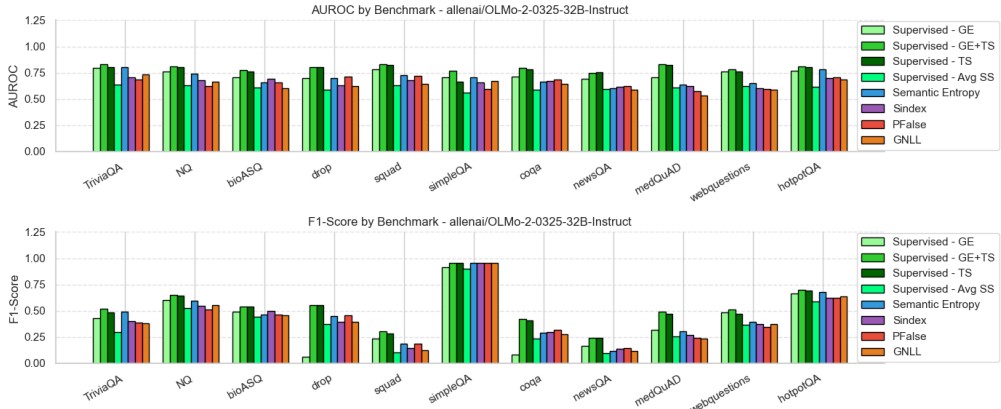

Figure 4: AUROC of supervised and unsupervised hallucination detector on *OLMo2-32B-Instruct*. Each group represents the heldout target dataset. Bars represent the method used with green bars showing different variants of supervised methods (differing by training data used) and remaining bars representing unsupervised methods.

**Broad domain shifts** While dataset-wise leave-one-out splits are a standard way to assess out of domain generalization, many of the evaluation datasets in our suite share substantial domain overlap with one another, for example NQ, TriviaQA both draw heavily from Wikipedia based sources, and thus we expect that domain transfer from them will be strong. To better disentangle this effect we curate GE sets based on 'broad' domains that are more dissimilar to one another, which we list in Table 1. These results are shown in Figure 5 which plot results of GE and GE+TS under different choices of the GE set. We only evaluate for two evaluation datasets, bioASQ and MedQuAD which have the most dissimilar domain in our set of evaluation benchmarks.

Across these settings we observe the same trend that using a GE set can result in a classifier with better performance than one that uses UQ metric based methods, this results stays consistent over several choices of GE sets and across models. We find that with the largest "Encyclopedic-Wiki" domain included, we did not see much variations in performance when including other domains.

**Analysis of Robustness Differences.** We further investigate the robustness gaps between the TS, GE and GE+TS setting using standard error decompositions under domain shift (Ben-David et al., 2010; Mansour et al., 2009). Our analysis suggests that TS training minimizes source (task-specific) error but amplifies divergences with other target (out-of-domain) datasets due to a heavier reliance on domain-specific features. In contrast, GE and GE+TS rely on more domain-invariant

features, reducing source-target divergence. Appendix H provides the full results and supporting
experiments based on domain classifications probes that empirically demonstrate these differences.

Overall, these results show that while the addition of a large heterogeneous general training mix
alone is insufficient to match the performance with target-specific data, it provides a strong foundation for supervised methods that surpasses unsupervised approaches on AUROC. In addition, we
find some complementary effect when these pretraining data are used together with target-specific
data.

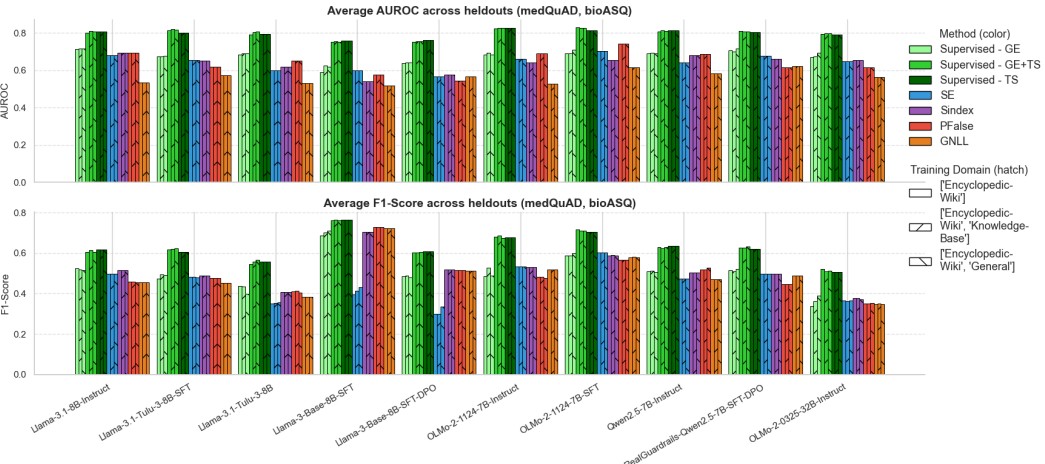

Figure 5: AUROC and F1-scores for aggregated over two target datasets - MedQuAD and bioASQ
under different choices of general data domain. Bars represent the method used with green bars
showing different variants of supervised methods (differing by training data used) and remaining
bars representing unsupervised methods. Hatching patterns on the bar denominate the choice of
general training domain when applicable.

### 4.3 SCALING LAWS: HOW MUCH IN DOMAIN DATA IS NEEDED TO ADAPT CLASSIFIER?

Our last investigation examines how much in-domain data labeled for hallucinations is needed to
train a classifier with adequate performance, and whether using a large general dataset can improve
data efficiency of this. Obtaining labeled hallucination data can be challenging, which makes a
general-domain data mix to supplement small in-domain labeled dataset for training classifiers. To
examine these effects, we plot learning curves (Perlich, 2011) under both the TS and GE+TS setting.
These curves plot the test-set performance as we vary the amount of task specific training data,
highlighting how quickly a classifier can learn to accurately detect hallucinations. For this setting,
we consider as in earlier only BioASQ and MedQuAD, which we deem as being most unlike the
other datasets. Figure 6 plots an example of these learning curves on `Llama 3.1 - Instruct`.
Generally, GE+TS dominates TS only in the low data regime of $< 1000$ examples, with TS-only
catching up soon after. This trend stays consistent across the 10 LLMs tested.

From these learning curves, we seek estimates of two quantities:

1. **Crossover Point**: At how many training examples does the TS-setting outperform the GE-
   only setting.
2. **Saturation Point**: At what sample count do we achieve 95% of detection performance in
   either the GE+TS or TS setting.

The crossover point between TS-only and GE-only marks the estimated amount of data where the
use of target specific data outperforms the use of the general set. This highlights a trade-off between
cost of annotation and model performance. A crossover point at a low number of samples suggests
that supervision from the target domain is highly valuable, whereas one at a high number of samples suggests that the general set itself might be the most practical choice unless a large and labeled
target/task-specific dataset is available. From the learning curves, we see that GE+TS curves almost

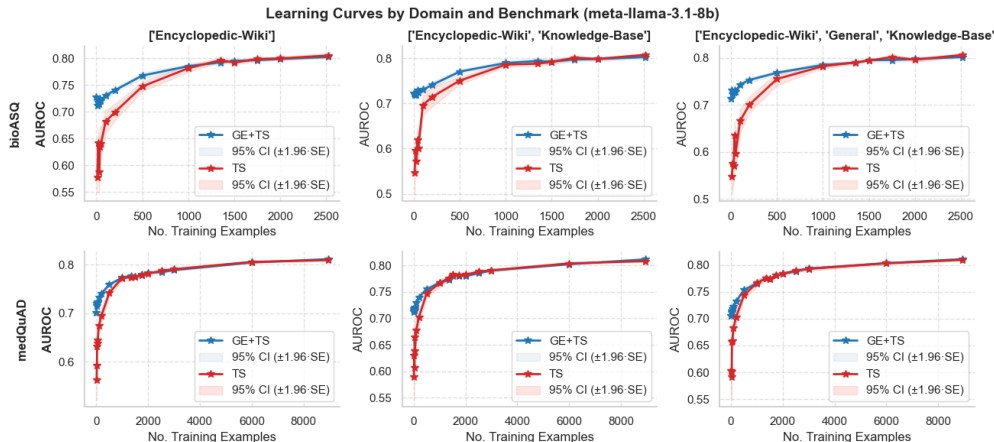

Figure 6: Learning Curves for `Llama-3.1-Instruct` under TS only and GE+TS for different amounts of target specific data. Top Row: Results under different GE set choices with BioASQ as target domains. Bottom Row: Results under different GE set choices with MedQuAD as target domains. Lines represent average AUROC value across 10 random seeds and shaded area represents $1.96 * SE$.

always lie above TS-only, indicating that GE+TS classifiers outperform or match of TS-only classifiers when given the same number of task-specific samples. Thus, we recommend the GE+TS setup, as it consistently provides equal or better performance regardless of the amount of task-specific data available. The saturation points give some guidance on how many labeled samples are needed in order to maximize the performance of a hallucination detection model.

Table 2: Performance on held-out datasets for cross-over experiments. Values are reported as mean (standard deviation)

| Held-out Data | N | Cross-over GE/TS | Saturation Point GE+TS | Saturation Point TS |
|---|---|---|---|---|
| BioASQ | 2520 | 372.9 (128.6) | 485.1 (201.9) | 820.7 (263.0) |
| MedQuAD | 8350 | 348.4 (149.6) | 1168.0 (306.0) | 1408.2 (712.4) |

Table 2 summarizes estimates for the both the crossover and saturation point. Across both heldout datasets, the expected cross-over between GE and TS occurs at roughly 310-350. Saturation point is reached relatively early for BioASQ, at 465 samples for GE+TS, and later for MedQuAD, at 1100 for GE+TS. We believe this is largely a function of the total number of samples we had for the experiment, in the case of medQuAD, the higher number of theoretical samples we had pushed the estimated maximal performance level higher, resulting in later saturation point. Consistently across both datasets, saturation point for GE+TS is much lower than that for TS only, 40.3% lower for bioASQ and 17.0% for MedQuAD, showing that use of the GE set improves data efficacy. Overall, these serve as a guidance for a practitioner deciding if labeling more data is worth it. For example, in the case of MedQuAD the expected saturation point of GE+TS at 1168 indicates that if one were to label an addition 7182 data points (8350-1168), they would only expect about a 5% increase in performance on AUROC.

## 5  CONCLUSION

In this paper we have presented a data-driven approach to make super hallucination detectors robust under domain shift. We first showed that supervised hallucination detection methods significantly outperforms unsupervised approaches in the in-domain setting, but that this advantage disappears in the out-domain setting, where unsupervised metric based approaches are comparatively more robust. We showed that the use of general, heterogeneous data that need not be in the same domain as the target domain can provide a useful foundation for training supervised classifiers, with such

classifiers generally surpassing unsupervised models even when data in the target domain is unavailable. Moreover our scaling experiments show that incorporating such general data improves data efficiency when combined with target specific data, as classifiers require fewer target-specific samples to achieve the same performance. These findings demonstrate that supervised UQ-based hallucination detection methods remain a valuable tool. Practitioners can apply classifiers trained on large general datasets and expect performance that exceeds unsupervised approaches, further, when available, incorporating target-specific data to these classifiers further improves performance, consistently outperforming unsupervised methods.

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

## A  UNSUPERVISED METHODS

SEMANTIC ENTROPY    Semantic entropy relies on assessing the consistency over multiple generations to a given reply. This is estimates uncertainty by measuring entropy of a semantic representation (might be wrong way to phrase it) across multiple sampled output For a single prompt $p$ we generate $S$ different responses $c_1, ..., c_S$. Semantic entropy clusters these responses into semantic equivalent clusters through the use of a Natural language inference model. Then using the semantic clusters we can compute semantic entropy by: formula

SINDEX    Similar to semantic entropy, Sindex also utilizes multiple generations, however instead of a NLI model to form clusters sindex uses sentence embeddings combined with a hierarchical clustering algorithm. Then an adjusted entropy is calculated by: ...

PTRUE    PTrue is an LLM as a judge approach that queries the generating LLM on whether a given statement is True or False, this is done by appending a question to the statement and measuring the generated token probability of the *true* token. Following Kadavath et al. (2022) we use a variation of PTrue where we pass in multiple candidate generations in the prompt as well as the main response (for which we score correctness). Additionally, for hallucination detection we actually need the probability inverse (1-pTrue) score, which we call pFalse.

**GNLL** GNLL is a likelihood-based score designed to estimate aleatoric uncertainty. Aichberger et al. (2024) show that under a 0-1 loss, the negative log-likelihood of the Maximum A Posterior completion is a good estimate (check this) of the aleatoric uncertainty. While this quantity is hard to identify due to computational intractability of the LLM generating space (find a citation for this) we can approximate this using either beam search decoding or greedy decoding. For this work we use the NLL of the greedy decoded sequence as GNLL.

## B  EVALUATION METRICS

**Threshold-Agnostic Metrics** Prior works on UQ and hallucination detection primarily evaluate performance using Area Under Receiver Operating Curve (AUROC) (Liu et al. (2024); Aichberger et al. (2024); Farquhar et al. (2024)). AUROC measures how well a score differentiates positive and negative examples. In our casem the score is either a UQ metric or the probability score generated by a supervised hallucination detector. As a threshold-agnostic metric, AUROC evaluates performance across all decision thresholds. Other options in this category are Area Under Precision Recall Curve (AUPRC) and Area Under Accuracy-Reject Curve (AUARC), but we primarily report performance using AUROC to keep consistent with previous work.

**Threshold-Aware Metrics** Many unsupervised metric-based methods output unbounded scores, this is the case for 3 out of 4 methods, SE, Sindex and GNLL. For them to be used practically we have to set thresholds for determining what are hallucinated responses. With these thresholds set, we can evaluate standard classification metrics such as accuracy, precision and recall. For this work, in order to optimize the threshold we use maximize F1-score on a separate validation set. To ensure fairness in comparison, we optimize thresholds for unsupervised methods on the training set, while for supervised detectors we rely on a separate validation subset.

## C  LLM AS A JUDGE

Here we give more details about the LLM-as-a-Judge Procedure. Broadly, we first evaluated GPT-4 as a judge model and found that its assessments were well aligned with human annotations, though on a relatively small sample. Given the cost of using GPT-4 at scale, we sought an open-source alternative with similar reliability. Among several candidates, we identified `Qwen3-14B` Yang et al. (2025), which exhibited the highest inter-annotator agreement with GPT-4. We therefore adopt `Qwen3-14B` as our primary judge model in this work (see Appendix X for details).

### C.1  LABELING

To generate labels for hallucination detection, we use a 3 class system:

- **0 - Non-Hallucinations** For generated responses deemed correct by the LLM judge
- **1 - Hallucinations** For generated responses deemed in-correct by the LLM judge
- **2 - Non Responses** For generated responses deemed as non-response by the LLM judge

We construct a small human-labeled set of 25 examples across datasets and model family, including brief annotations of why they are labeled the way they are. These examples are used in a few-shot prompt (25-shot) to guide the LLM judge in labeling the full set of responses, this prompt is given in Figure 7.

### C.2  OPEN-SOURCE MODELS

Our initial efforts used mainly `gpt-4o` as a judge, but we found this prohibitively expensive for the number of experiments we wanted to do, thus we sought an open-source alternative that would perform as well. We mainly tested models from the Qwen3 family, named the 14B transformer and the 30B MoE model. We found Cohen's Kappa to be higher for the 14B transformer as shown in Table **??**. We also annotate by hand a sample of a 100 model completions and compare that with our models, which is shown in Table 3, for this sample GPT-4o and Qwen3-14B perform nearly

---

**LLM-as-a-Judge Prompt**

You are an expert evaluator for question-answering systems. You will be assessing the quality of answers to a given question.

**Task:** Determine if the candidate answer contains the correct factual information to answer the question.

**Rules:** - Respond with `0` if the candidate answer is equivalent in meaning to any reference answer (synonyms/context OK), or if it contains the specific correct answer. - Respond with `1` if the candidate contains wrong facts, repeats the question, provides no answer, has the wrong entity, or differs from all reference answers. - Respond with `2` if the model does not give an answer, asks a clarifying question, or refuses to answer. - Accept additional context around correct answers. - Accept geographic or temporal equivalents. - Reject if the core answer is missing or incorrect.

**Instructions:** Reference answers may be in a numbered list. Score `0` if the candidate matches any reference answer.

**Evaluation Examples:**

- Q: What movie starred Tom Cruise? Ref: Top Gun Candidate: "Top Gun starring Tom Cruise" → 0

- Q: What movie starred Tom Cruise? Ref: Top Gun Candidate: "Brad Pitt was in Top Gun" → 1

- Q: What's the name of Mob's brother's Spanish VA in Mob Psycho 100? Ref: Javier Olguín Candidate: "I do not have information on the Spanish voice actor..." → 2

- Q: What is the major difficulty in carrying out the plan? Ref: Improving the relationship between Taiwan and the mainland Candidate: "I'd be happy to help you identify potential difficulties..." → 2

---

Figure 7: LLM-as-a-Judge prompt used for labeling responses from different datasets. Only 4 few-shot examples are shown here due to space constraints, but for actual applications we use a 25-shot example

identical. Further both have high accuracy and inter annotator agreement, further validating our choice to use Qwen3-14B

| Model | Cohen's Kappa v Human | Accuracy vs Human |
|---|---|---|
| GPT-4o | 0.84 | 92% |
| Qwen3-14B | 0.85 | 93% |

Table 3: Inter-annotator agreement between LLM-as-a-judge models and human raters.

| Model | Cohen's Kappa v GPT-4o | Accuracy vs GPT-4o |
|---|---|---|
| GPT-4o | 1.00 | 100% |
| Qwen3-14B | 0.756 | 85.4% |
| GPT-4o-Mini | 0.808 | 89.4% |
| Qwen3-A3B30B | 0.659 | 82.2% |

Table 4: Inter-annotator agreement between LLM-as-a-judge models and human raters.

Table 5: Models evaluated in our study.

| Hugging Face Model name | Model family | Size | Training stage |
|---|---|---|---|
| meta-llama/Llama-3.1-8B-Instruct | Llama-3.1 | 8B | Instruct |
| allenai/Llama-3.1-Tulu-3-8B-SFT | Llama-3.1 | 8B | SFT |
| allenai/Llama-3.1-Tulu-3-8B | Llama-3.1 | 8B | Instruct |
| princeton-nlp/Llama-3-Base-8B-SFT | Llama-3 | 8B | SFT |
| princeton-nlp/Llama-3-Base-8B-SFT-DPO | Llama-3 | 8B | SFT + DPO |
| allenai/OLMo-2-1124-7B-Instruct | OLMo-2 | 7B | Instruct |
| allenai/OLMo-2-1124-7B-SFT | OLMo-2 | 7B | SFT |
| Qwen/Qwen2.5-7B-Instruct | Qwen2.5 | 7B | Instruct |
| normster/RealGuardrails-Qwen2.5-7B-SFT-DPO | Qwen2.5 | 7B | SFT + DPO |
| allenai/OLMo-2-0325-32B-Instruct | OLMo-2 | 32B | Instruct |

# D  LLMS TESTED

# E  SUPERVISED TRAINING DETAILS

Here we furnish addition details on the training procedure and model use.

## E.1  DATASET

Table 6 gives additional information on dataset sizes and the splits to create the train test and validation sets per evaluation benchmark.

Table 6: Benchmarking Datasets used in our study.

| Benchmark | Domain | Total Size | Train Size | Test Size |
|---|---|---|---|---|
| TriviaQA | Encyclopedic | 100K | 10k | 10k |
| NQ | Encyclopedic | 120K | 10k | 10k |
| bioASQ | Biology | 50K | 10k | 10k |
| CoQA | Conversational | 120K | 10k | 10k |
| DROP | Reasoning-heavy | 96K | 10k | 10k |
| HotpotQA | Encyclopedic | 113K | 10k | 10k |
| MedQuAD | Medical | 50K | 10k | 10k |
| NewsQA | News | 120K | 10k | 10k |
| SimpleQA | KB QA | 100K | 10k | 10k |
| SQuAD | Encyclopedic | 100K | 10k | 10k |
| WebQuestions | KB QA | 6K | 10k | 10k |
| OpenLLM | Various | 22K | 10k | 10k |

## E.2  SUPERVISED HALLUCINATION CLASSIFIER

The supervised learning model used as the classifier is a Random Forest (Breiman, 2001). We did not do extensive hyperparameter tuning per generating LLM and dataset due to compute constraints, instead opting for a setting of 100 trees and with remaining settings being the default in scikit-learn. We also included an additional dimensionality reduction step since for many of our experiments we have the case where the dimensionality of the features exceeded the number of training examples. This dimensionality reduction is carried out using Singular Value Decomposition (SVD), for which we use the implementation in scikit-learn. We set a fix dimensionality of 300 after SVD, which is then used in the Random Forest. Lastly as a pre-processing step before SVD we apply standard scaling to the raw features.

**Features**  Following Liu et al. (2024) we use activations from the middle and last layer of the model. For each of these layers we take the activation value corresponding to both the last token and

the prompt, this creates a large feature space as we take a total of 4 activations per input, for example if the model's hidden size is 4096 then the size of the features corresponding to this activation is 16384 (4*4096). In addition we explored the use of several probability based features but find that they did not impact performance much and omitted them.

## F  RQ1 ADDITIONAL FIGURES

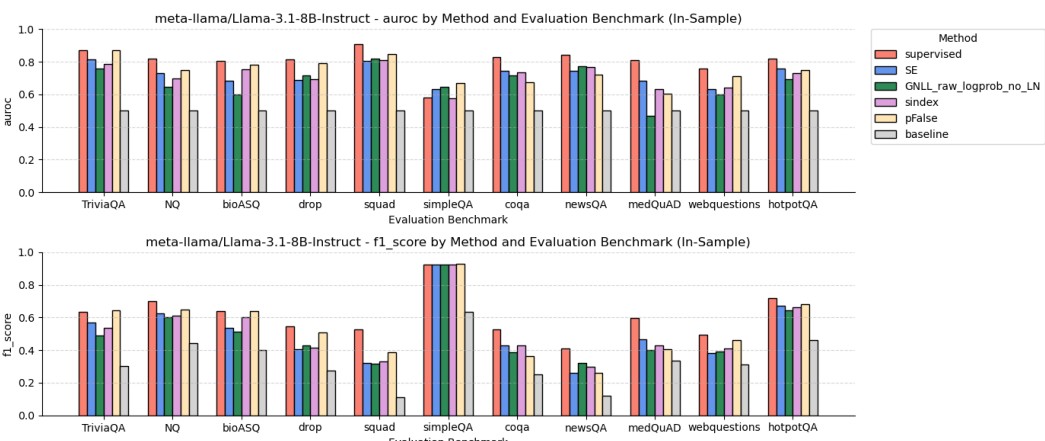

Figure 8: In-domains evaluations of "Llama-3.1-Instruct" across 11 different datasets. Supervised methods plotted along with 4 unsupervised methods and one random classifier baseline. Top Row: AUROC, Bototm Row: F1-Score

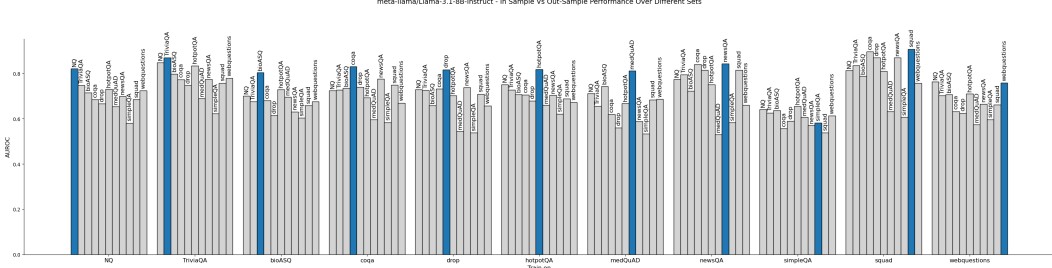

Figure 9: In-domains evaluations for various models macro-averaged across 11 different datasets. In order to benchmark supervised methods against unsupervised methods we pick the top performing unsupervised method for macro-averaging. Top Row: AUROC, Bottom Row: F1-Score

## G  RQ2 ADDITIONAL FIGURES

## H  DOMAIN SPECIFICITY OF VARIOUS CLASSIFIERS

### H.1  DOMAIN SHIFT AND ERROR DECOMPOSITION

Here we provide further analysis on the different behaviors of GE, TS and GE+TS based hallucination detectors. We first consider the hypothetical error decomposition under domain shift (Ben-David et al., 2010; Mansour et al., 2009). Let $D_s$ and $D_t$ denote the source and target distributions. Then where $\epsilon_t(h)$ denotes the target error of hypothesis $h \in \mathcal{H}$, we have:

$$\epsilon_t(h) \leq \epsilon_s(h) + \frac{1}{2}d_{\mathcal{H}\Delta\mathcal{H}}(D_s, D_t) + \lambda \qquad (2)$$

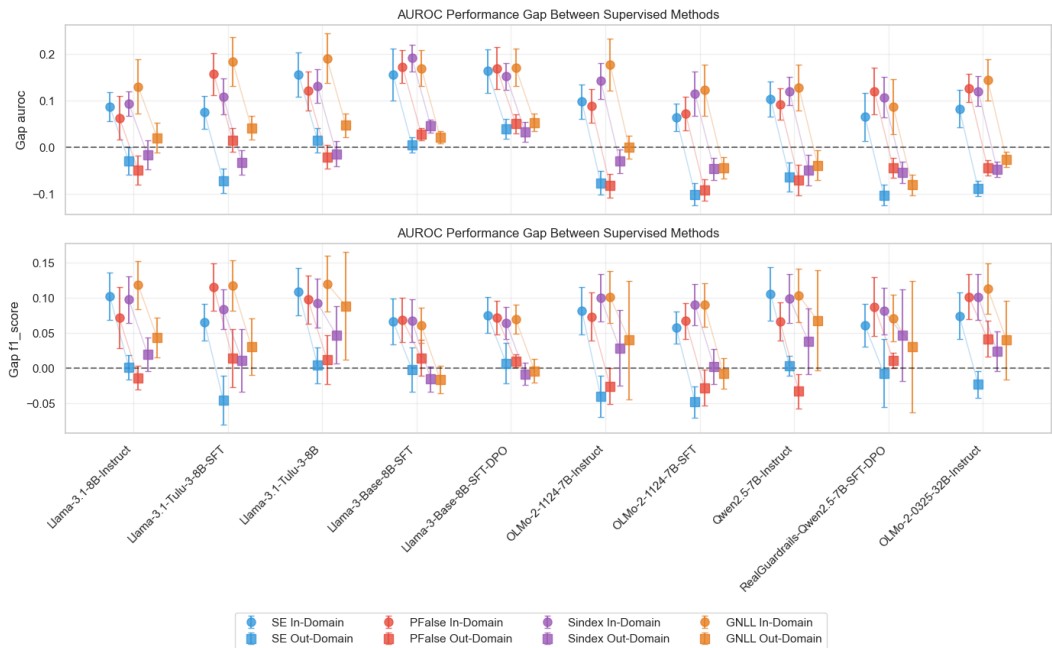

Figure 10: Difference plots comparing unsupervised methods to supervised methods. Heights of marker correspond to the performance difference between supervised methods and unsupervised methods. Top: AUROC, Bottom: F1-Score. Positive values indicate that supervised methods performance bettter than unsupervised method. Circles represent in-domain and squares represent out-of-domain. Error Bars correspond to 1.96 * Standard Error calculated over scores from aggregated source-target pairs.

The first term $\epsilon_s(h)$ denotes the error on the source domain. $\frac{1}{2}d_{\mathcal{H}\Delta\mathcal{H}}(D_s, D_t)$ is a divergence term that measures how different source and target distributions are in the feature space induced by $h$. The third term $\lambda$ is a hypothesis mismatch term which captures aggregate performance of the best hypothesis $h^* \in \mathcal{H}$ in both source and target domain. Interpreting our three training regimes TS,GE and GE+TS within this framework we hypothesize that:

- **TS Trained Classifiers** minimize $\epsilon_s(h)$, but have high divergence $d_{\mathcal{H}\Delta\mathcal{H}}(D_s, D_t)$ due to the encoding or use of domain-specific features.
- **GE Trained Classifiers** trained on heterogeneous datasets in contrast, should create a feature representation that reduces $d_{\mathcal{H}\Delta\mathcal{H}}(D_s, D_t)$, although at the cost of increasing source specific error $\epsilon_s(h)$ on any specific dataset.
- **GE+TS Trained Classifiers** are likely to have a favorable balance of both terms.

We have seen that TS trained hallucination detectors generally have much higher performance than GE trained variants, validating that these models have a lower source specific error. To explain why TS trained classifiers tend to fail out of domain, week seek to validate whether they indeed rely more heavily on domain-specific features.

## H.2 DOMAIN CLASSIFICATION PROBES AND EXPERIMENTAL SETUP

To obtain a proxy that is compatible with the discrete feature sets used in our random forest classifiers, we exploit the Gini importance based feature ranking produced during training to train domain classification probes. Our hypothesis is that classifiers trained on task-specific data will utilize features that better encode domain-specific artifacts, resulting in better performance on the domain identification task over GE and GE+TS feature sets.

For each training regime (TS, GE, GE+TS) we extract the top 10% of features in the hallucination detector model. We then train a downstream classifier whose goal is not to detect hallucinations, but

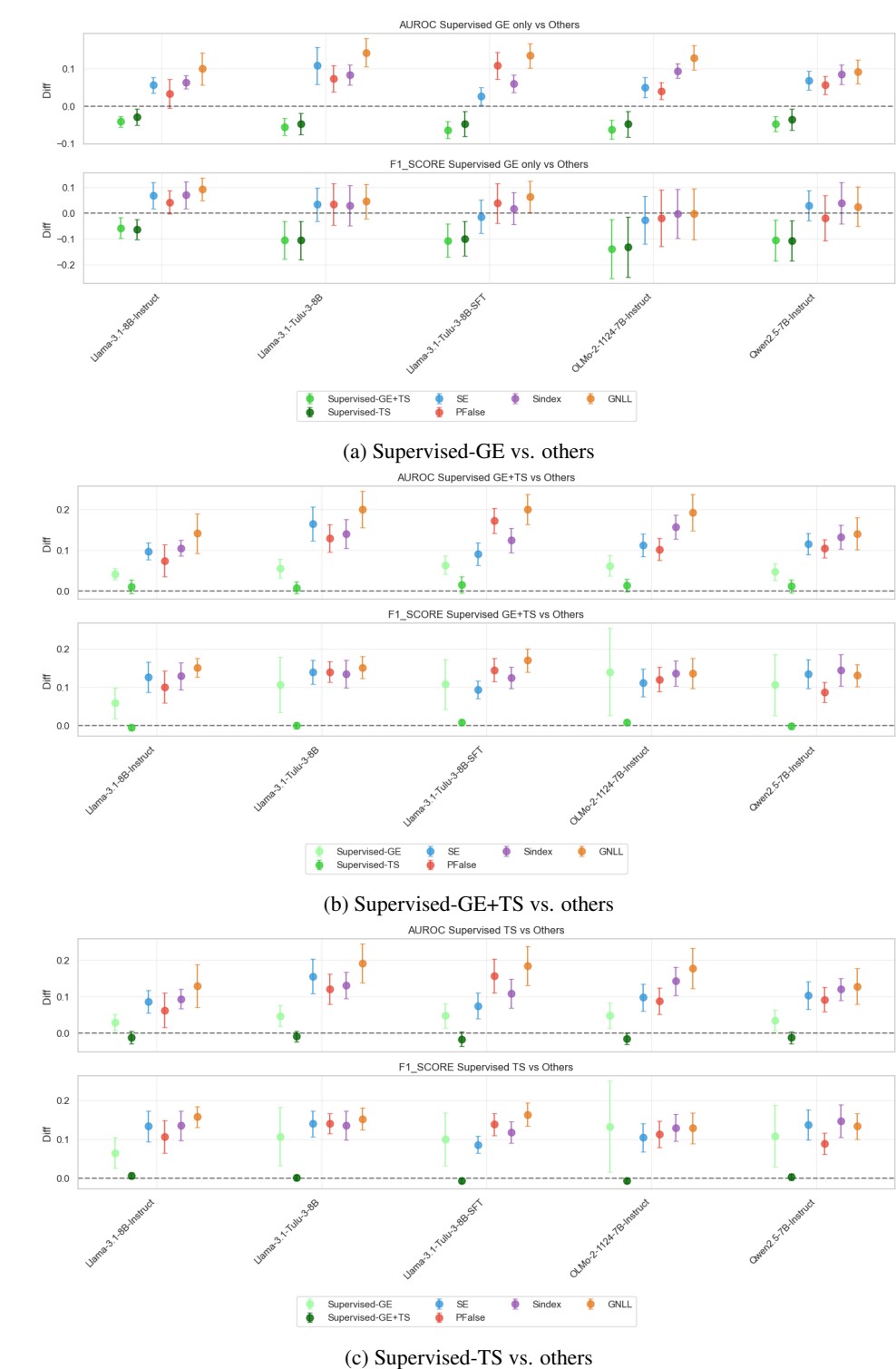

(a) Supervised-GE vs. others

(b) Supervised-GE+TS vs. others

(c) Supervised-TS vs. others

Figure 11: Difference plots comparing the performance of (a) Supervised-GE, (b) Supervised-GE+TS, and (c) Supervised-TS against all other methods.

rather to predict from which data set a sample came from. This is done by training a binary domain classification probe for every task-specific dataset $D_{TS}$, where we set the positive class as samples

from $D_{TS}$ and the negative class as samples from $D_{GE}$. To evaluate we look at the F1-score relative to the F1-score obtained by a domain classifier trained on all available features.

### H.3   RESULTS

Figure 12 displays the relative F1-Score for 5 task-specific datasets bioASQ, MedQUAD, SimpleQA, CoQA and NewsQA, the GE set chosen is the 'encyclopedic-wiki' set which consists of datasets with encyclopedic like content (Table 1). The aggregated results show that **TS trained classifiers tend to prioritize features which are domain specific**, generally achieving higher relative F1-Score over GE and GE+TS selected feature sets. Figure 13 displays the same results but disaggregated to their individual LLMs. We see that the trend is consistent, hPTolding for almost all tested datasets in 10 of the 12 models.

These results provide empirical evidence explaining the domain shift problem under TS regime, and why GE and GE+TS training helps mitigate this. TS-only models may have low bias on source domain, but experience large domain shifts. In contrast GE only models may have higher bias as they miss task-specific nuances experience a smaller domain shift effect. Lastly we have seen that GE+TS balances both the bias and domain shift term.

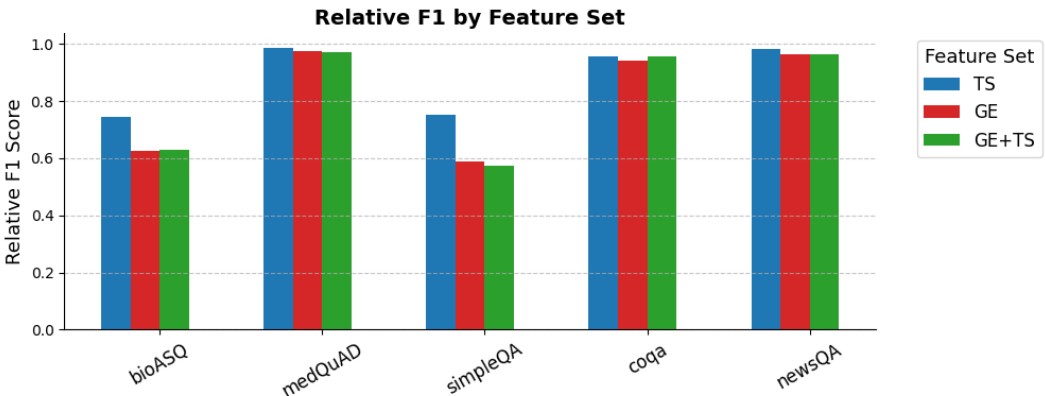

Figure 12: Relative F1-score of domain classification probes trained under 3 feature sets, TS, GE, GE+TS, scores are aggregated across 12 LLMs

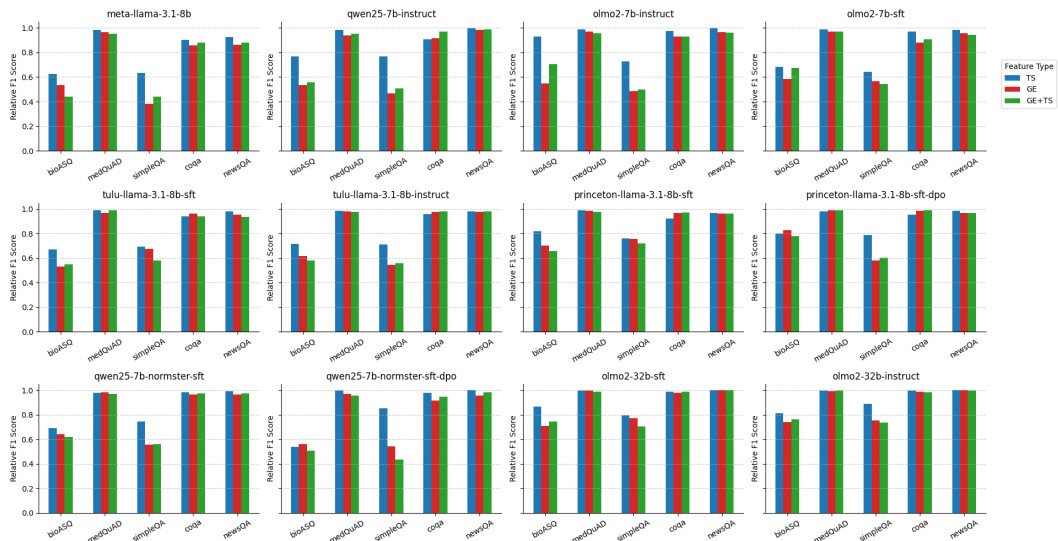

Figure 13: Relative F1-score of domain classification probes trained under 3 feature sets, TS, GE, GE+TS, each plot shows results for one of 12 tested LLMs

