# OpenReview forum: "Fortifying Hallucination Detection to Out-of-Domain Data"
_ICLR.cc/2026/Conference — ICLR 2026 Conference Withdrawn Submission_

### Official Review · Reviewer_4hng · 2025-10-31

**Soundness:** 2
**Presentation:** 2
**Contribution:** 1
**Rating:** 2
**Confidence:** 4

**Summary:**

The paper investigates the domain shift problem in supervised hallucination detection for Large Language Models (LLMs), where supervised methods demonstrate strong in-domain performance but degrade significantly under distribution shift. The papers propose a data-centric solution involving training supervised detectors on a diverse "general training mix" to improve out-of-domain robustness. They evaluate their approach across 697K QA samples from 12 datasets and 10 different LLMs, reporting an average improvement of +7.25% in out-of-domain performance and demonstrating that their approach can reduce task-specific data requirements by up to 40.3%.

**Strengths:**

1. Comprehensive empirical evaluation: The study spans a substantial experimental scope with 697K samples across 12 diverse QA datasets and 10 different LLMs, providing extensive coverage of domain shifts and model architectures.
2. Clear problem identification: The work systematically documents and quantifies the generalization gap between in-domain and out-of-domain performance for supervised hallucination detection, which is a legitimate practical concern.
3. Practical applicability: The proposed solution directly addresses a real-world deployment challenge where hallucination detection systems must operate across diverse domains.
4. Thorough benchmarking: The comparison against state-of-the-art unsupervised methods (Semantic Entropy, SIndex, GNLL, PTrue) across multiple evaluation metrics (AUROC, F1-score) provides a comprehensive baseline.

**Weaknesses:**

1. Limited novelty in methodology: The core contribution—using diverse training data for domain generalization—is a well-established technique in machine learning, particularly in domain adaptation literature. The application to hallucination detection, while practically motivated, does not introduce novel methodological innovations or theoretical insights.
2. Insufficient principled analysis: The paper lacks deep theoretical analysis of why diverse training improves out-of-domain performance in the specific context of uncertainty-based hallucination detection. The underlying mechanisms remain under-explored.
3. Methodological superficiality: The approach relies primarily on empirical observations without providing theoretical foundations or formal analysis of the generalization properties of the proposed method.

**Questions:**

1. What theoretical insights can you provide about why diverse training improves hallucination detection under domain shift? How do these insights extend beyond simple domain adaptation principles?
2. How do you reconcile the apparent contradiction between the reported benefits and the observation of "negative transfer" when relying on heterogeneous general data?
3. What specific components of the general training mix are most responsible for the observed improvements? Have you conducted systematic ablation studies to identify these?

---

> ### Author Response · Authors · 2025-11-25
>
> We greatly appreciate the reviewer’s feedback and comments. To respond to the concerns and questions.
>
> *Insufficient principled analysis: The paper lacks deep theoretical analysis of why diverse training improves out-of-domain performance in the specific context of uncertainty-based hallucination detection. The underlying mechanisms remain under-explored.*
>
> *Methodological superficiality: The approach relies primarily on empirical observations without providing theoretical foundations or formal analysis of the generalization properties of the proposed method.*
>
> *What theoretical insights can you provide about why diverse training improves hallucination detection under domain shift? How do these insights extend beyond simple domain adaptation principles?*
>
> We have included additional experiments and analysis to gain further insights into why training on heterogenous dataworks. These are included in appendix H, with a summary in the main body. Broadly, we consider the hypothetical error decomposition under domain shift  (Ben-David et al., 2010; Mansour et al., 2009), and obtain proxy estimates of the divergence term between source and target domains. We then analyse how these estimates vary under the TS, GE and GE+TS regime change.
>
> In summary, our analysis shows that TS-trained hallucination detectors minimize source error but rely more heavily on domain-specific features, which amplifies source-target divergence. In contrast, the GE and GE+TS setting results in classifiers that utilize more domain-invariant features, reducing source-target divergence. We validate this by using domain-classification probes which predict source labels of input prompts. We find that probes trained with TS selected features are generally more predictive of source labels than those trained with GE or GE+TS selected features, this result is observed over 5 datasets and 12 different LLMs. This combined evidence explains the observed robustness gaps of TS only model and why GE+TS fixes this, TS models tend to overfit to domain-specific features, while GE and GE+TS do not, balancing better source accuracy and domain shift, thus yielding improved out-of-domain performance.
>
> [1] Shai Ben-David, John Blitzer, Koby Crammer, Alex Kulesza, Fernando Pereira, and Jennifer Wort-man Vaughan. A theory of learning from different domains. Machine learning, 79(1):151–175, 2010.
>
> [2] Yishay Mansour, Mehryar Mohri, and Afshin Rostamizadeh. Domain adaptation: Learning bounds and algorithms. arXiv preprint arXiv:0902.3430, 2009

---

### Official Review · Reviewer_cMrM · 2025-11-01

**Soundness:** 3
**Presentation:** 1
**Contribution:** 2
**Rating:** 4
**Confidence:** 3

**Summary:**

This paper introduces a data-driven approach to enhance the generalization of supervised hallucination detection methods. Through extensive experiments on 11 open-source QA datasets, the authors demonstrate that incorporating general training data improves the performance of supervised methods on out-of-domain datasets.

**Strengths:**

1. **Comprehensive Experiments**: Experiments span 11 open-source QA datasets and 10 different LLMs. Compare supervised and unsupervised methods in both the in-domain and out-of-domain settings.

2. Introduces a data-driven method to improve the cross-domain generalization of supervised hallucination detectors.

**Weaknesses:**

1. **Limited Scope**: This paper only conduct experiments on QA dataset for white(grey)-box hallucination detection method.

2. **Hallucination Labeling**: The labeling fully relies on LLM judges, and potential biases or errors from the judge model could affect the results.

3. **Limited Insight**: The findings are not particularly insightful; the observation that general data can improve out-of-domain performance and enhance data efficiency is largely expected.

4. **Undefined Terms**: PT (Line 321) and FT (Line) 367 is introduced without prior explanation and not defined later.

5. **Missing Appendix Links**: The main text does not include clickable links to the appendices (e.g., Appendix C and D), making it hard to check important details.

**Questions:**

## Questions
I am confused about the exact size of the GE dataset and the mixing ratios of the different datasets used to construct it. I expect the training dataset to be large, but the experiments in Section 4.3 suggest that only a few thousand examples are used.

## Suggestions
1. The paragraph titled “Evaluation Dataset” is not entirely accurate, as these datasets are later split into training and test sets for the leave-one-out experiments.
2. Line 736 references a table that is missing.

---

> ### Author Response · Authors · 2025-11-25
>
> We greatly appreciate the reviewer’s feedback and comments. To respond to the concerns and questions.
>
> *Limited Scope: This paper only conduct experiments on QA dataset for white(grey)-box hallucination detection method.*
>
> We agree that extending to black-box methods would be valuable, however with the scale of our evaluations (>600K) samples, generating responses from a proprietary model would be prohibitively expensive, this was also why we chose to use the open-source Qwen3 14B model for our LLM as a judge. We chose to focus on QA tasks since those are generative (vs MCQ based) as well as easily verifiable.
>
> *Hallucination Labeling: The labeling fully relies on LLM judges, and potential biases or errors from the judge model could affect the results.*
>
> We appreciate the concern regarding potential biases with LLM as a judge. 1. We found high levels of agreement between human annotated examples (see Appendix C). 2. Human annotation at scale is not feasible, with >600K prompts across >10 llm models, the number of examples is too high for human labelling to be considered. 3. LLM as a judge represents the highest performing verification method amongst automated approaches, previous work has shown that relying on simpler approaches like string matching popular in earlier hallucination detection work can lead to inaccuracies and the wrong model being chosen.
>
>
> *Limited Insight: The findings are not particularly insightful; the observation that general data can improve out-of-domain performance and enhance data efficiency is largely expected*
>
> We respectfully disagree that our findings offer limited insights. While we do confirm that general data improves out of domain performance in the supervised setting, our results reveal several additional novel findings that challenge assumptions in previous work .
>
> 1.  Prior work (Liu et al., 2024) reports that hallucination detectors already generalize well across domains, suggesting no need for training the general+task specific setting. Our results challenge this assumption by showing that detectors trained on narrow sets of data will likely fail, and that this can be easily mitigated by including GE data.
>
> 2. Our work also shows that with these out-of-domain issues addressed, supervised detectors out-perform unsupervised approaches (which are often favoured for their domain agnosticism). This thus also provides clarity to practitioners on what types of methods to use for hallucination detection.
>
> Additionally, to gain further insights into why training on heterogeneous data works, we have included an additional study in appendix H as well as a summary in the main body on the role that general data and task-specific data play and how these shape the trained classifier.
>
> In summary, our analysis shows that TS-trained hallucination detectors minimize source error but rely more heavily on domain-specific features, which amplifies source-target divergence. In contrast, the GE and GE+TS setting results in classifiers that utilize more domain-invariant features, reducing source-target divergence. We validate this by using domain-classification probes which predict source labels of input prompts. We find that probes trained with TS selected features are generally more predictive of source labels than those trained with GE or GE+TS selected features, this result is observed over 5 datasets and 12 different LLMs. This combined evidence explains the observed robustness gaps of TS only model and why GE+TS fixes this, TS models tend to overfit to domain-specific features, while GE and GE+TS do not, balancing better source accuracy and domain shift, thus yielding improved out-of-domain performance.
>
>
> *Undefined Terms: PT (Line 321) and FT (Line) 367 is introduced without prior explanation and not defined later.*
>
> Thanks for pointing this out! We have corrected this in the paper.
>
> *Missing Appendix Links: The main text does not include clickable links to the appendices (e.g., Appendix C and D), making it hard to check important details.*
>
> Thank you for this suggestion. We have corrected this in the paper.
>
> *I am confused about the exact size of the GE dataset and the mixing ratios of the different datasets used to construct it. I expect the training dataset to be large, but the experiments in Section 4.3 suggest that only a few thousand examples are used.*
>
> The size of the GE set is generally large (section 2) this would encompass roughly 500k samples depending which are heldout. In section 3, we consider more ‘domain-oriented’ GE sets, the sizes of these vary with the largest (encyclopedic + wiki) being about 300k examples. We will reword the section to make this more explicit.
>
> [1] Linyu Liu, Yu Pan, Xiaocheng Li, and Guanting Chen. Uncertainty estimation and quantification for llms: A simple supervised approach. arXiv preprint arXiv:2404.15993, 2024.

---

### Official Review · Reviewer_HEmX · 2025-11-05

**Soundness:** 3
**Presentation:** 3
**Contribution:** 3
**Rating:** 6
**Confidence:** 4

**Summary:**

This paper studies domain adaptation for LLM-based QA models. Many empirical findings are stated, including 1) supervised > unsupervised in in-domain settings, 2) training with general (GE) data improves OOD, and 3) GE + task-specific (TS) data leads to lower annotation costs.

There are extensive experiments and results to justify the claim. The evaluation uses the LLM-as-judge pipeline as the evaluator for accuracy, and includes F1 and AUROC as metrics. The reliability of the judge itself is also shown. Additionally, the findings align with much existing work (different tasks), making the conclusion credible.

Overall, I believe this is a solid piece of research, and its conclusion is convincing in the given setting: LLM only, without RAG or external knowledge.

**Strengths:**

1. Easy to follow and clearly stated. The results are presented in clear, concise language without overemphasis.

2. The reliability of the LLM-as-judge is evaluated, and its limitation is discussed for transparency.

3. The white/grey-box pipeline experiments show the correlations between the model's internal states and errors (hallucinations), and I think the design is novel.

**Weaknesses:**

1. Despite this paper's findings in the QA task, similar conclusions have been stated in other tasks [1,2]. The impression of "contributing new knowledge"  (ICLR reviewing guideline) is not very significant, but this is a subjective view.

2. The results are presented as bar charts, but they are too dense with information. Using tables while highlighting the difference between average unsupervised methods vs. supervised methods should be better.


References

[1] Zihan Liu, Yan Xu, Tiezheng Yu, Wenliang Dai, Ziwei Ji, Samuel Cahyawijaya, Andrea Madotto, Pascale Fung. CrossNER: Evaluating Cross‑Domain Named Entity Recognition. AAAI 2021
[2] Jindong Wang, Cuiling Lan, Chang Liu, Yidong Ouyang, Tao Qin, Wang Lu, Yiqiang Chen, Wenjun Zeng, Philip S. Yu. Generalizing to Unseen Domains: A Survey on Domain Generalization. IEEE TKDE 2022

**Questions:**

What does task-specific domain data do? Suppose such data does not cover the knowledge for a test case. How could it help reduce the hallucination problem of LLM? Does it help its reasoning pattern or lower confidence for uncertain questions?

---

> ### Author Response · Authors · 2025-11-25
>
> We greatly appreciate the reviewers feedback and comments. To respond to the concerns and questions:
>
> *Despite this paper's findings in the QA task, similar conclusions have been stated in other tasks [1,2]. The impression of "contributing new knowledge" (ICLR reviewing guideline) is not very significant, but this is a subjective view.*
>
> While we agree that prior work has observed that training on heterogenous data can improve out of domain generalization, to the best of our knowledge this has not been demonstrated in the context of supervised hallucination detection. Existing studies (Liu et al., 2024) in fact report that supervised hallucination detectors already generalize well out of domain. We believe that these conclusions are partly due to differences in evaluation settings, with earlier works relying on simpler correctness functions (ROUGE/BLEU) and more limited domain coverage. In contrast, our study uses LLM as a judge for correctness across a more diverse set of datasets and models, enabling us to uncover these robustness gaps that were previously unobserved. Our results show that supervised hallucination detections trained on task-specific data do not generalize reliably, and that standard methods in domain generalization can overcome this. Thus, our work makes a novel empirical contribution in this space.
>
> *The results are presented as bar charts, but they are too dense with information. Using tables while highlighting the difference between average unsupervised methods vs. supervised methods should be better.*
>
> We appreciate the reviewers suggestion regarding presentation of results. While we found bar charts useful for simultaneously showing performance of the models as well as differences between in-domain and out of domain, we agree that they may appear dense. To address this we will include numerical results in tables in the appendix.
>
>
> *What does task-specific domain data do? Suppose such data does not cover the knowledge for a test case. How could it help reduce the hallucination problem of LLM? Does it help its reasoning pattern or lower confidence for uncertain questions?*
>
> Task-specific data (in-domain data) enables supervised hallucination detectors to perform well on inputs that are similar to the task-specific distribution. From a standard machine learning perspective additional in-domain data should help minimize empirical risk reducing error on similar unseen test points. Even if data does not cover exactly the questions (inputs) seen at test time, as long as the linguistic and structural characteristics of the test case question remain similar to the task-specific training data the model should generalize.
>
> In order to highlight more rigorously the connections between training on task-specific and generation data and how these affect downstream performance both in-domain and out of domain, have added additional experiments as well as a theoretical framework in Annex H, with a summary incorporated into section 3 (both highlighted in blue for easier identification). In summary, our analysis shows that TS-trained hallucination detectors minimize source error but rely more heavily on domain-specific features, which amplifies source-target divergence. In contrast, the GE and GE+TS setting results in classifiers that utilize more domain-invariant features, reducing source-target divergence. We validate this by using domain-classification probes which predict source labels of input prompts. We find that probes trained with TS selected features are generally more predictive of source labels than those trained with GE or GE+TS selected features, this result is observed over 5 datasets and 12 different LLMs. This combined evidence explains the observed robustness gaps of TS only model and why GE+TS fixes this, TS models tend to overfit to domain-specific features, while GE and GE+TS do not, balancing better source accuracy and domain shift, thus yielding improved out-of-domain performance.
>
> [1] Linyu Liu, Yu Pan, Xiaocheng Li, and Guanting Chen. Uncertainty estimation and quantification
> for llms: A simple supervised approach. arXiv preprint arXiv:2404.15993, 2024.

---

### Author Response · Authors · 2025-12-02
**Summary Of Discussions**

To the AC, thank you for taking the time to review our submission in this unusual review cycle. In this comment we would like to briefly summarize some of the key points brought up by the reviewers and to highlight our corresponding clarifications and addition experiments.

**On Novelty**

Reviewer HEmX noted:

*Despite this paper's findings in the QA task, similar conclusions have been stated in other tasks [1,2]. The impression of "contributing new knowledge" (ICLR reviewing guideline) is not very significant, but this is a subjective view.*

This is similar to reviewer 4hng who noted:

*Limited novelty in methodology: The core contribution—using diverse training data for domain generalization—is a well-established technique in machine learning, particularly in domain adaptation literature. The application to hallucination detection, while practically motivated, does not introduce novel methodological innovations or theoretical insights.*

For these we responded:

While we agree that prior work has observed that training on heterogenous data can improve out of domain generalization, to the best of our knowledge this has not been demonstrated in the context of supervised hallucination detection. Existing studies (Liu et al., 2024) in fact report that supervised hallucination detectors already generalize well out of domain. We believe that these conclusions are partly due to differences in evaluation settings, with earlier works relying on simpler correctness functions (ROUGE/BLEU) and more limited domain coverage. In contrast, our study uses LLM as a judge for correctness across a more diverse set of datasets and models, enabling us to uncover these robustness gaps that were previously unobserved. Our results show that supervised hallucination detections trained on task-specific data do not generalize reliably, and that standard methods in domain generalization can overcome this. Thus, our work makes a novel empirical contribution in this space.

**On Lack of theoretical insight into why general data aids in domain generalization**

Reviewer cMrM noted:

*Limited Insight: The findings are not particularly insightful; the observation that general data can improve out-of-domain performance and enhance data efficiency is largely expected.*

This is similar to reviewer 4hng who asked:

*What theoretical insights can you provide about why diverse training improves hallucination detection under domain shift? How do these insights extend beyond simple domain adaptation principles?*

To address these comments:

We  include additional analysis on GE data's effect on domain adaptation through the lens of error decomposition under domain shift.
We have included additional experiments and analysis to gain further insights into why training on heterogenous dataworks. These are included in appendix H, with a summary in the main body. Broadly, we consider the hypothetical error decomposition under domain shift (Ben-David et al., 2010; Mansour et al., 2009), and obtain proxy estimates of the divergence term between source and target domains. We then analyse how these estimates vary under the TS, GE and GE+TS regime change.

In summary, our analysis shows that TS-trained hallucination detectors minimize source error but rely more heavily on domain-specific features, which amplifies source-target divergence. In contrast, the GE and GE+TS setting results in classifiers that utilize more domain-invariant features, reducing source-target divergence. We validate this by using domain-classification probes which predict source labels of input prompts. We find that probes trained with TS selected features are generally more predictive of source labels than those trained with GE or GE+TS selected features, this result is observed over 5 datasets and 12 different LLMs. This combined evidence explains the observed robustness gaps of TS only model and why GE+TS fixes this, TS models tend to overfit to domain-specific features, while GE and GE+TS do not, balancing better source accuracy and domain shift, thus yielding improved out-of-domain performance.

**Others**


Additionally, we corrected several formatting and labeling mistakes. In addition some sections have had the wording changed slightly to make things clearer.

[1] Linyu Liu, Yu Pan, Xiaocheng Li, and Guanting Chen. Uncertainty estimation and quantification for llms: A simple supervised approach. arXiv preprint arXiv:2404.15993, 2024.

[2] Shai Ben-David, John Blitzer, Koby Crammer, Alex Kulesza, Fernando Pereira, and Jennifer Wort-man Vaughan. A theory of learning from different domains. Machine learning, 79(1):151–175, 2010.

[3] Yishay Mansour, Mehryar Mohri, and Afshin Rostamizadeh. Domain adaptation: Learning bounds and algorithms. arXiv preprint arXiv:0902.3430, 2009

---

### Note · Authors · 2026-01-05

**Comment:**

We are withdrawing this submission due to an internal decision by the authors.

**Withdrawal Confirmation:**

I have read and agree with the venue's withdrawal policy on behalf of myself and my co-authors.